# Research on the Hydrological Variation Law of the Dawen River, a Tributary of the Lower Yellow River

Yan Li [1,†], Long Zhao [2,†], Zhe Zhang [1], Jianxin Li [2], Lei Hou [2,*], Jingqiang Liu [2,*] and Yibing Wang [2]

1 Hydrological Center of Tai'an City, Tai'an 271001, China; liyan_taswj@shandong.cn (Y.L.); zhangzhe_taswj@shandong.cn (Z.Z.)
2 College of Water Conservancy and Civil Engineering, Shandong Agricultural University, Tai'an 271018, China; zlllong0421@gmail.com (L.Z.); lijianxinqwe@gmail.com (J.L.); wybing926@gmail.com (Y.W.)
* Correspondence: houl@sdau.edu.cn (L.H.); jqliu@sdau.edu.cn (J.L.)
† These authors contributed equally to this work.

**Abstract:** The natural runoff mechanism of the Dawen River, the main tributary of the lower Yellow River, has been stressed in recent years as a result of human activity, and the hydrological situation has changed dramatically. In this paper, various hydrological statistical methods such as the Mann–Kendall nonparametric test, cumulative anomaly, ordered clustering, sliding T test, and rainfall–runoff double-cumulative curve were used to study the evolution characteristics of hydrological factors in Dawen River. The result revealed that the rainfall and runoff of the Dawen River decreased overall from 1956 to 2016, but the downward trend was not clear, and the runoff variance was high, with 1978 as the variation point. The IHA/RVA and PCA were used to comprehensively evaluate the hydrological variability of the Dawen River, and nine representative indicators were screened out. The overall change was 58%, which is mild, and the difference in hydrological change between the IHA index system and the PCA index system was just 7%, which was predictable. The hydrological situation of the Dawen River has undergone huge changes, and there has been a serious dry-off phenomenon since 1978. The biology, habitat, and structure of the Dawen River have all been irreversibly impacted by changes in its hydrological regime. Furthermore, the key influencing aspect of hydrological variation is the vast building of water conservation schemes. The findings could serve as a theoretical foundation for integrated water resource management and ecological conservation.

**Keywords:** Dawen River; hydrological statistical analysis; IHA-RVA; principal component analysis; hydrological variation and its affecting factors





## 1. Introduction

Water availability for agriculture, industry, and cities is important for long-term societal growth [1]. Because of social progress, the public demand for water has increased year by year, and shortages have become widespread [2]. Wang et al. [3] studied runoff changes in 1321 watersheds around the world, and the results showed that global runoff has decreased by an average of 11.9 mm/yr in recent decades. Over the period 1948–2004, Su et al. [4] investigated the long-term flow trends of 916 of the world's biggest sea-guzzling rivers: 503 had reduced flow, while 408 rivers had an increase. There are more rivers with less flow than with more. The positive flow tendency is more prevalent at high latitudes, while the negative flow trend is more prevalent at low latitudes. River flows in Europe exhibit a south (dry) to north (wet) differential, which is widening [5]. Water scarcity in the Mediterranean region is becoming more acute as a result of climate change, and the water shortage situation in Africa will worsen as the population grows, precipitation and runoff decrease, and evapotranspiration rises [6,7]. In addition, precipitation in India is on the decline [8], a trend that is having a detrimental impact on the region's weather and environment. Moreover, India's political, economic, cultural, and social structures will be altered. The level of Lancang–Mekong River, responsible for 60% of the global

freshwater flow [9], fell to its lowest level in 50 years in 2010 due to an unusual change in climate, a condition that affects Southeast Asian countries' long-term socioeconomic development [10].

Changes in hydrological conditions jeopardize the virtuous circle of social economic development and environmental protection and has an impact on human production and existence; however, rainfall and runoff trends can be analyzed to offer useful information on hydrological variables, water resource planning, and management systems. Understanding the performance and features of long-term hydrological and climatic variables is critical for developing countries' socioeconomic development and for the efficient management and usage of basin water resources [7]. Although China has a vast population, its per capita water resources are just a fifth of the global average. More than half of China's provinces are experiencing water shortages due to the country's unique geographical and climatic characteristics. Because China's water resources are unevenly distributed [11], hydrological factors should receive more attention. It is important to diagnose variation in hydrological and meteorological elements to understand the causes. This has important theoretical significance and practical value for the rational development of basin water resource management and the mastery of the basin water cycle.

In the 1990s, the runoff of rivers in northern China changed dramatically [12]. From 1995 to 2018, the annual runoff of the Haihe, Liaohe, and Yellow River basins decreased significantly, while the annual runoff of the Yangtze, Huaihe, and Pearl River basins in South China decreased slightly. Tian et al. [13] used the IHA–RVA method to analyze the hydrological variation of the Wuding River in northwest China from 1960 to 2016. According to the study, the entire hydrological change of the Wuding River was 69%, and land-use changes and the building of water conservation projects in the region had a significant impact on the hydrological condition. Liu et al. [2] investigated the runoff trend in the Yellow River's middle and upper sections, as well as the elements that influenced it. The results showed that the runoff is declining sharply over the last 50 years primarily due to climate change. Reduced runoff in the Yellow River's middle and lower reaches was mostly due to the installation of measures and water conservation projects, as well as water transfer across river basins. Xu et al. [14] investigated temporal and regional variations in Yellow River hydrology over the last 60 years and found that human activity, reservoir construction and control, and water resource exploitation all had a significant impact on hydrological changes.

Richter et al. [15] proposed indicators of hydrologic alteration (IHA) to assess hydrological changes. The IHA comprised 32 indicators, but the number of indicators and their grouping were adjusted to 33 in 1998, divided into five indicator groups [16]. This method is now acknowledged as the most systematic set of indicators. Richter et al. [17] proposed the range of variability approach (RVA) for a better quantitative explanation of the changes, and it has subsequently become widely used. For example, Fantin-Cruz et al. [18] evaluated the impact of water diversion hydropower facilities on the hydrological status of the Correntes River in Brazil; Mwedzi et al. [19] analyzed the degree of change in hydrological indicators in different sections of the Manyame River Basin in Zimbabwe after the dam was built. However, the numerous IHA indicators have a relatively strong association [20], and indicators with a correlation will repeat the description of the degree of change, resulting in a small or large evaluation [21]. Many studies seek to explore and eliminate redundancies between IHA indicators for which the autecology matrix (AM) [22], PCA [23–25], and genetic programming (GP) [26] methods are common. The PCA method is widely used because of its objective and reasonable determination of the index weight and standardized calculation process, which is simple to implement on the computer. Identifying the mutation points of hydrological sequences is the foundation for studying hydrological variation laws. The following are all commonly used to identify mutation points in hydrological sequences: the cumulative anomaly method [27–29], Mann–Kendall test (MK) [30–32], ordered cluster analysis [33], double cumulative-curve method [34,35], Pettitt method [36–38], moving t test [39,40], Lee–Heghinian method [13,41], and Hurst

method [42,43]. In the study of hydrological variation points, a variety of mutation testing methods are typically used [44–47].

The Dawen River is the lower Yellow River's greatest tributary. Downstream of the Dawen River is Dongping Lake, and Dongping Lake undertakes the tasks of flood storage and water transfer from the South to the North. In addition, the Dawen River Basin is home to almost 5.09 million people and 5.4 million hectares of agricultural land. The Dawen River is the main source of water for agricultural irrigation, production, and domestic use in the region. The discharge of the Dawen River has changed dramatically in recent decades as a result of human disturbance [48]. Flow cutoff was first observed in May 1978, according to the measured daily runoff at the Daicun Dam Hydrological Station. Between 1979 and 2016, there were 97 months with no runoff, accounting for 21.3% of the total. It not only has a major impact on water for industrial and agricultural production and people's domestic use, but also poses a huge threat to the regional ecological environment. Therefore, the study of hydrological variation of Dawen River is of great significance to the development of water resources and the protection of ecological environment.

The main purpose of this study is to explore the evolution characteristics of precipitation and runoff in the Dawen River. At the same time, the combination of PCA and IHA was used to study the hydrological variation law of the Dawen River and to analyze the change of the hydrological situation of the Dawen River and its impact on the river ecological environment. In addition, the main influencing factors of the hydrological variability of the Dawen River were further clarified.

## 2. Material and Methods

### 2.1. Study Area

The Dawen River (35°42′ N–36°36′ N, 116° E–118° E) is the lower Yellow River's greatest tributary [48]. The river is located in Tai'an City, Shandong Province, and flows from east to west. The Dawen River has two tributaries: the Yingwen and Chaiwen rivers. The object of this work is the Daicun Dam Hydrological Station's control basin (known as the upper and middle reaches of the Dawen River in this paper), which covers 9098 km$^2$. In the east, the terrain is high, while in the west it is flat. Mountainous areas make up 31%, and hilly areas 37%. The average annual precipitation in the Dawen River Basin is 709 mm (1956–2016), with June–September accounting for almost 75%. Drought prevails in other months. Spring precipitation is lower in the basin, temperatures are high, and there are spring droughts. Summer is hot and rainy; autumn is dry, with a significant drop in temperature; and winter is cold and dry, with little precipitation. The climate in the basin is semi-humid. The geographical map of Dawen River Basin is shown in Figure 1.

### 2.2. Data Sources

The data come from the *Yellow River Basin Hydrology Yearbook* and are based on annual rainfall data from 26 rainfall stations in the Dawen River Basin from and daily measured runoff data from Daicun Dam Hydrological Station from 1956 to 2016. The dataset for the Digital Elevation Model (DEM) in this paper is ASTER GDEM 30M, which can be found on the Geospatial Data Cloud Platform (https://www.gscloud.cn/ (accessed on 1 March 2022)).

### 2.3. Hydrological Statistical Analysis Methods

(1)  Mann–Kendall text

For trend analyses of hydrological and meteorological variables, the Mann–Kendall (MK) statistical method [49,50] is frequently used [51]. It is a non-parametric test for detecting trends in time series data that does not require the data to follow a normal distribution [52].

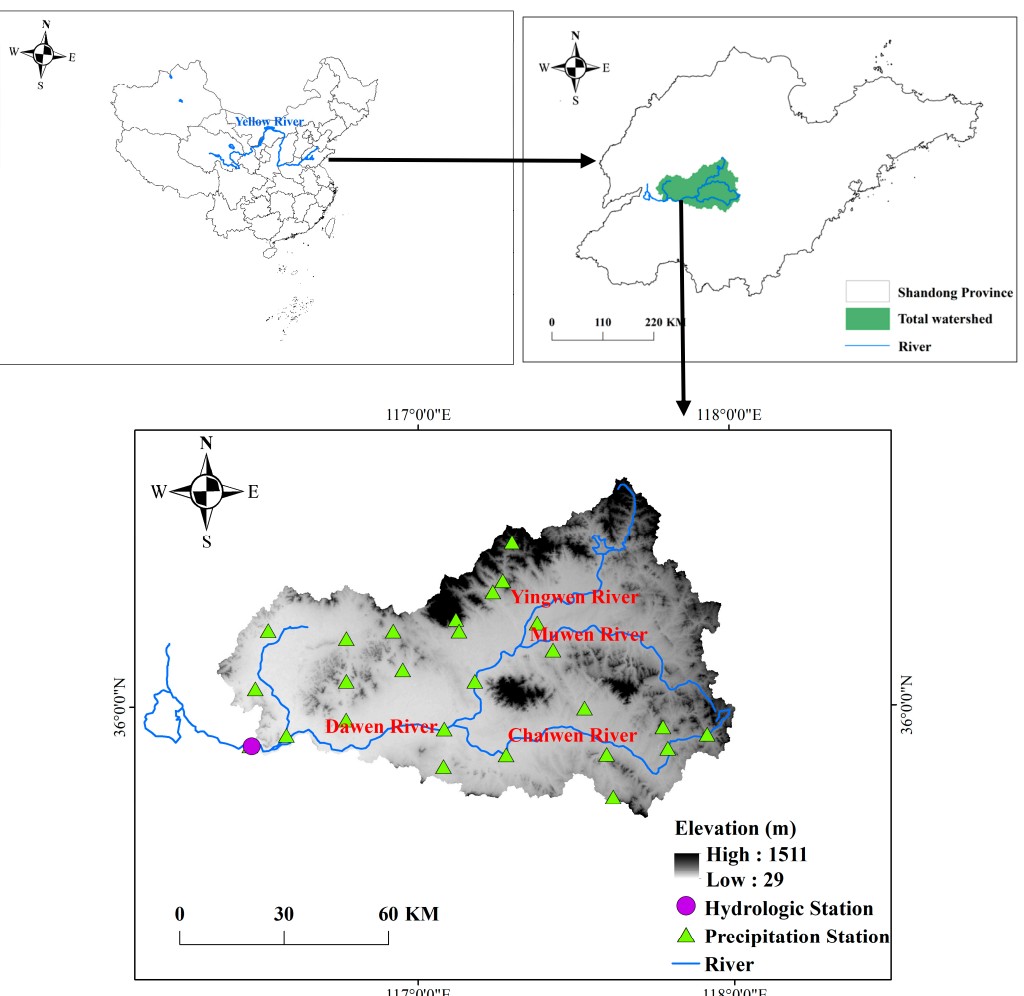

**Figure 1.** Geographical map of the study area.

The specific method is as follows:

For time series data $Y = \{y_1, y_2, y_3, \cdots, y_n\}$, define its statistics:

$$S = \sum_{i=1}^{n-1} \sum_{j=i+1}^{n} \text{sgn}\left(y_j - y_i\right) \tag{1}$$

In the formula:

$$\text{sgn}\left(y_j - y_i\right) = \begin{cases} 1 & \text{if } y_j > y_i \\ 0 & \text{if } y_j = y_i \\ -1 & \text{if } y_j < y_i \end{cases}$$

The random number sequence $S_a (a = 1, 2, 3, \cdots, n)$ obeys a normal distribution, and its variance is

$$V(s) = \frac{n(n-1)(2n+5)}{18} \tag{2}$$

Set random sequence S independent and define the $M - K$ statistics:

$$Z_{MK} = \begin{cases} \frac{S-1}{\sqrt{V(s)}} & S > 0 \\ 0 & S = 0 \\ \frac{S+1}{\sqrt{V(s)}} & S < 0 \end{cases} \tag{3}$$

When the level of explicitness is $\beta = 0.05$, $\left|Z_{1-\frac{2}{\beta}}\right| = 1.96$; when it is $\beta = 0.01$, $\left|Z_{1-\frac{2}{\beta}}\right| = 2.58$. To begin, the random number series is assumed to have a non-changing trend. When $|Z_{MK}| > \left|Z_{1-\frac{2}{\beta}}\right|$, reject the null hypothesis and the trend becomes more obvious; otherwise, it is not.

Mutation refers to the rapid transition from one state to another state [53].

The basic principle of the Mann–Kendall mutation test is as follows: Construct the order column according to the time series data; calculate the statistics UF and UB according to the order column; after a given significance level, the intersection of the statistics UF and UB within the critical value is the sequence mutation point [53].

If the value of UF or UB is greater than zero, it indicates that the series is in an upward trend, and if it is less than zero, it indicates a downward trend. When they exceed the critical straight line, it indicates a significant upward or downward trend.

(2)  Cumulative anomaly method

In hydrological research, the cumulative anomaly method [54] is a statistical tool for analyzing the mutation of hydrological and meteorological data. The following is the procedure for calculating the cumulative anomaly value: first, the difference between the long-series data and the average value is computed. Then, the cumulative anomaly value accumulates year by year according to the time sequence. For sequences x, the cumulative anomaly is $\sum_{i=1}^{t}(x_i - \bar{x})$. The following are the normalized cumulative anomalies:

$$K = \frac{\sum_{i=1}^{t}(x_i - \bar{x})}{\bar{x}}, t = 1, 2, \ldots n \tag{4}$$

If the value K progressively grows, it means that each point in the long-series data is larger than the average and trending upward.

(3)  Rainfall–runoff double-cumulative curve

In the statistical study of hydrology, the rainfall–runoff double mass analysis [55] is commonly employed. The underlying premise is that rain and runoff build over the same amount of time. The abscissa represents cumulative rainfall, whereas the ordinate represents cumulative runoff. The rainfall–runoff double-cumulative curve can be used to examine runoff trend change characteristics. If the slope of the rainfall–runoff double-cumulative curve changes, it indicates that the runoff has mutated on this time scale, with the mutation year being the evident slope shift. The mutation points of the precipitation–runoff double-cumulative curve is the starting point for studying precipitation and runoff variations over time. The following is the formula:

$$X'_i = \sum_{i=1}^{N} X_i \tag{5}$$

$$Y'_i = \sum_{i=1}^{N} Y_i \tag{6}$$

In the formula: $X'_i$ is the annual accumulated precipitation (mm); N is the column length; i is the time series; $X_i$ is the precipitation in year i (mm); $Y'_i$ is the annual cumulative streamflow (m$^3$/s); $Y_i$ is the streamflow in year i (m$^3$/s).

*2.4. IHA/RVA*

The Indicators of the Hydrologic Alteration (IHA) method total 33, and they are used to evaluate the characteristics of hydrological changes. These indicators are classified into five ecologically significant indicators: discharge magnitude, time, frequency, duration, and rate of change [15,56]. The grouping of indicators and their corresponding ecosystem impacts

are shown in Table 1. The IHA can analyze trends over a single time period or compare two different ones in the hydrological record. If the hydrological system undergoes abrupt changes, such as while constructing a dam, a two-phase analysis should be performed. A single-period study should be employed for hydrological systems that have undergone long-term human change.

**Table 1.** The conventional 33 IHA indicators.

| IHA Parameter Group | Hydrologic Parameters | Ecosystem Influences |
|---|---|---|
| Group 1 Magnitude of monthly water conditions | Median streamflow for each month (Subtotal 12 parameters) | • Habitat availability for aquatic organisms <br> • Soil moisture availability for plants <br> • Availability of water for terrestrial animals <br> • Availability of food/cover for fur-bearing mammals <br> • Reliability of water supplies for terrestrial animals <br> • Access by predators to nesting sites <br> • Influences water temperature, oxygen levels, photosynthesis in water column |
| Group 2 Magnitude of annual extreme discharge events with different durations | 1-day minimum <br> 3-day minimum <br> 7-day minimum <br> 30-day minimum <br> 90-day minimum <br> 1-day maximum <br> 3-day maximum <br> 7-day maximum <br> 30-day maximum <br> 90-day maximum <br> Zero streamflow days <br> Base streamflow index <br> (Subtotal 12 parameters) | • Balance of competitive, ruderal, and stress-tolerant organisms <br> • Creation of sites for plant colonization <br> • Structuring of aquatic ecosystems by abiotic vs. biotic factors <br> • Structuring of river channel morphology and physical habitat conditions <br> • Soil moisture stress in plants <br> • Dehydration in animals <br> • Anaerobic stress in plants <br> • Duration of stressful conditions such as low oxygen and concentrated chemicals in aquatic environments <br> • Distribution of plant communities in lakes, ponds, floodplains <br> • Duration of high flows for waste disposal, aeration of spawning beds in channel sediments |
| Group 3 Timing of annual extreme water conditions | Minimum streamflow date <br> Maximum streamflow date <br> (Subtotal 2 parameters) | • Compatibility with life cycles of organisms <br> • Predictability/avoidability of stress for organisms <br> • Access to special habitats during reproduction or to avoid predation <br> • Spawning cues for migratory fish <br> • Evolution of life history strategies, behavioral mechanisms |
| Group 4 Frequency and duration of high and low pulses | Low pulse count <br> Low pulse duration <br> High pulse count <br> High pulse duration <br> (Subtotal 4 parameters) | • Frequency and magnitude of soil moisture stress for plants <br> • Frequency and duration of anaerobic stress for plants <br> • Availability of floodplain habitats for aquatic organisms <br> • Nutrient and organic matter exchanges between river and floodplain <br> • Soil mineral availability <br> • Access for waterbirds to feed, rest, and reproduce <br> • Influences bedload transport, channel sediment textures, and duration of substrate disturbance (high pulses) |
| Group 5 Rate and frequency of water condition changes | Rise rate <br> Fall rate <br> Number of reversals <br> (Subtotal 3 parameters) | • Drought stress on plants (falling levels) <br> • Entrapment of organisms on islands, floodplains (rising levels) <br> • Desiccation stress on low-mobility streamedge (varial zone) organisms |

The Range of Variability Approach (RVA) is a method for quantifying the degree of hydrological variation in rivers on the basis of the IHA method [16,17]. The method details are as follows: assess the degree of hydrological variability of rivers by analyzing the degree of change in each indicator before and after the mutation point.

The formula of each indicator hydrological change degree $A_i$ is

$$A_i = \left| \frac{e_i - e_0}{e_0} \right| \times 100\% \tag{7}$$

In the formula: $e_i$ is the number of years for which the i index value is within 25–75% after mutation; $e_0 = \gamma e_t$; $\gamma$ is the proportion of the indicator before the impact that falls within the ecological target threshold (50% in this paper); $e_t$ represents the number of years since the mutation.

After mutation, the total hydrological alteration degree $A_0$ is

$$A_0 = \sqrt{\left( \frac{1}{33} \sum_{I=1}^{33} A_i^2 \right)} \tag{8}$$

To express the degree of change in ecological and hydrological indicators qualitatively, when $0\% \leq |A_0| \leq 33\%$, it is low; when $33\% < |A_0| \leq 67\%$, it is moderate; when $67\% < |A_0| \leq 100\%$, it is high.

### 2.5. Principal Component Analysis (PCA)

The principal component analysis is a multi-variable data analysis method established by Harold Hotelling in 1933 to extract the most important information from data sets and produce new orthogonal variables. These variables are linear combinations of the original variables [57], generated by using the original data set's covariance matrix to generate a new set of axes in the direction of the data set's largest variance, referred to as principal components (PCs) [51].

The goal of this method is to extract the most important information from the data set, compress the size of the data set by reducing the dimension, and ensure that too much information will not be lost. Therefore, this method is often used in research to screen the indicators that can best summarize the global information.

## 3. Results

### 3.1. Evolution Characteristics of Hydrological Factors in Dawen River

3.1.1. Trend Analysis

The precipitation in the upper and middle reaches of the Dawen River (above Daicun Dam hydrological station) showed a decreasing trend year by year, with a decrease of 0.819 mm/yr as shown (Figure 2a). The data used were the average precipitation from 1956 to 2016 at 26 rainfall stations. In this paper, the Mann–Kendall nonparametric test was used to analyze the trend, and the statistic Z was −0.786. Since $|Z_{MK}| < 1.96$ ($\beta < 0.05$ level of significance), the downward trend was not significant. The streamflow trend is shown in Figure 2b, showing a downward trend. Its trend was $8 \times 10^6$ m$^3$/a, and the Mann−Kendall test statistic value was 1.458. Similarly, its downward trend was not significant. By making that comparison, it was discovered that streamflow had a more pronounced downward tendency than precipitation, with Hurst values of 0.71 and 0.54, indicating that runoff was more volatile.

As a result, while streamflow's decreasing trend was not considerable, its volatility was, and this had an influence on ecology, production, and daily life.

The MK test was used to investigate the evolution characteristics of rainfall using average annual precipitation from 26 rainfall stations in the basin from 1956 to 2016, the results are shown in Figure 3. Only one site in this watershed showed a substantial negative tendency from the overall trend, and the tendencies of other stations were not clear (failed to pass the 90 percent significance test). Overall precipitation was decreasing across the entire basin, but this trend was not very significant. The explanation for the minor shift in precipitation is that climate change has not been drastic in recent decades, resulting in minor variations in precipitation. Climate, topography, and other factors all played a role.

The rainfall distribution was quite varied from east to west, with rainfall increasing and then decreasing. Rainfall in high−altitude locations was heavier than in low-altitude areas.

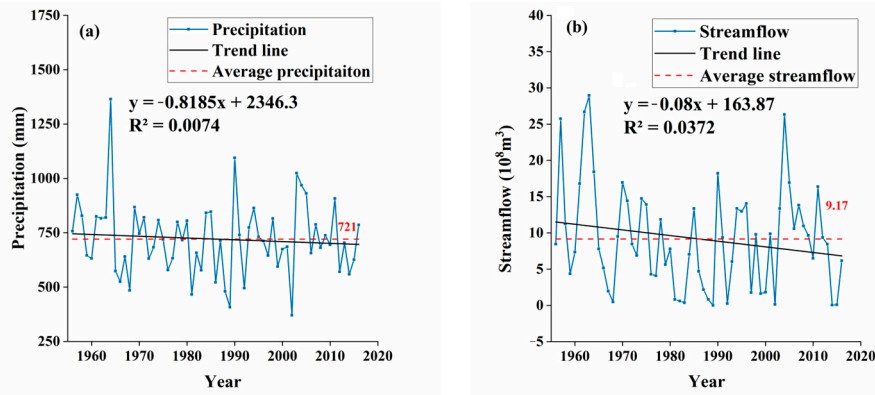

**Figure 2.** Precipitation (**a**) and streamflow (**b**) trend map of Dawen River Daicun Dam Basin.

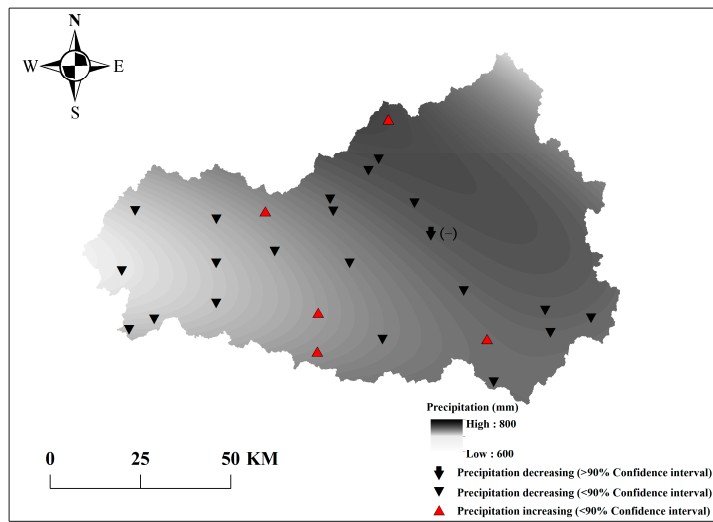

**Figure 3.** Spatial distribution of the annual precipitation trend from the MK test in the middle and upper reaches of the Dawen River.

### 3.1.2. Analysis of Mutation

The MK mutation test, cumulative anomaly method, and ordered clustering method were used to conduct the mutation analysis on the measured annual runoff from 1956 to 2016 in the middle and upper reaches of the Dawen River. The results are shown in Table 2. The comprehensive analysis showed that the variation points from 1956 to 2016 at Daicun Dam Hydrological Station were 1964 and 1978. The rainfall in 1964 was 1365 mm, which is significantly greater than the annual average rainfall of 721 mm. The runoff variation in 1964 was caused by a sharp increase in rainfall. When compared to the precipitation-measured runoff double-accumulative curve (Figure 4), the slope of the precipitation–runoff double-accumulative curve dropped after 1978, showing that human activity had a greater influence on runoff. The hydrological variation law of the Dawen River was researched with 1978 as the variation point to better understand the impact of human activity on the hydrological situation of the Dawen River.

**Table 2.** Result of the mutation test of annual measured runoff of Dawen River.

| Hydrologic Station | Mutation Year | | | Variation Point |
|---|---|---|---|---|
| | **Mann–Kendall Test** | **Cumulative Anomaly Method** | **Moving *t* Test** | |
| Daicun Dam | 1964, 1968, 1978, 1993 | 1964, 1978 | 1964, 1969, 1975, 1978, 2002 | 1978 |

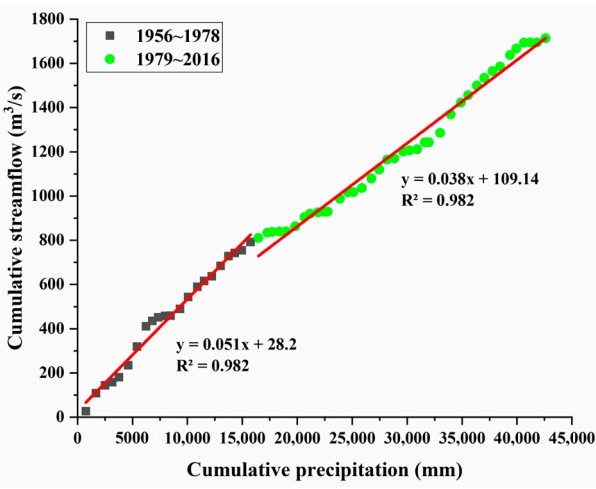

**Figure 4.** Precipitation–streamflow double-accumulative curve at Daicun Dam Hydrological Station.

*3.2. The Law of Hydrological Variation of Dawen River*

3.2.1. Degree of Hydrological Change of Dawen River

The research period of the hydrological variation law was separated into two periods on the basis of the year of runoff variation: before variation (1956–1978) and after (1979–2016). We analyzed the changes in 33 indicators using the IHA/RVA method, then calculate the degree of change for each indicator; the results are shown in Table 3. These hydrological indicators changed significantly after the change, and the overall hydrological change rate was 54%. Nine indicators had significant changes, twelve had moderate changes, and twelve indicators had minor changes. There were some disparities in the degree of hydrological change among the groups, as shown in Table 4. The second group of indicators (annual extreme flow intensity and duration) and the fifth (flow change rate and frequency) both had relatively high degrees of change, indicating that human activity had a greater impact on river and bias river runoff towards even distribution. The third group of indicators (the period when the extreme annual flow occurs) showed little variation, demonstrating that the annual rainfall distribution before and after the variation changed little.

**Table 3.** Changes of hydrological indicators in the basin.

| Group | Serial Number | IHA Indicators | Pre-Impact Period | Post-Impact Period | RVA Boundaries Low | High | Hydrologic Alteration Numerical Value (%) | Degree of Change |
|---|---|---|---|---|---|---|---|---|
| Group 1 | 1 | Median streamflow in October | 12.3 | 2.87 | 2.78 | 21.4 | −28% | L |
| | 2 | Median streamflow in November | 13.2 | 4.47 | 7.78 | 18.1 | −54% | M |
| | 3 | Median streamflow in December | 8.16 | 2.87 | 5.18 | 12.6 | −61% | M |
| | 4 | Median streamflow in January | 7.62 | 5.00 | 4.51 | 9.76 | −8% | L |
| | 5 | Median streamflow in February | 4.23 | 3.32 | 2.69 | 8.30 | 5% | L |
| | 6 | Median streamflow in March | 2.32 | 1.41 | 1.11 | 3.31 | −48% | M |
| | 7 | Median streamflow in April | 1.15 | 0.221 | 0.627 | 1.98 | −61% | M |
| | 8 | Median streamflow in May | 1.63 | 0.000 | 0.454 | 2.24 | −80% | H |
| | 9 | Median streamflow in June | 0.881 | 0.000 | 0.294 | 1.63 | −67% | H |
| | 10 | Median streamflow in July | 76.4 | 11.5 | 32.1 | 118 | −15% | L |
| | 11 | Median streamflow in August | 61.8 | 40.3 | 32.9 | 104 | −8% | L |
| | 12 | Median streamflow in September | 23.4 | 10.1 | 15.7 | 60.4 | −41% | M |
| Group 2 | 13 | 1-day minimum | 0.322 | 0.000 | 0.146 | 0.689 | −87% | H |
| | 14 | 3-day minimum | 0.352 | 0.000 | 0.148 | 0.803 | −87% | H |
| | 15 | 7-day minimum | 0.373 | 0.000 | 0.221 | 1.14 | −87% | H |
| | 16 | 30-day minimum | 0.614 | 0.000 | 0.296 | 1.42 | −87% | H |
| | 17 | 90-day minimum | 1.28 | 0.245 | 0.582 | 2.15 | −54% | M |
| | 18 | 1-day maximum | 963 | 416 | 603 | 1254 | −28% | L |
| | 19 | 3-day maximum | 669 | 328 | 414 | 801 | −34% | M |
| | 20 | 7-day maximum | 455 | 251 | 311 | 539 | −41% | M |
| | 21 | 30-day maximum | 207 | 141 | 161 | 295 | −28% | L |
| | 22 | 90-day maximum | 101 | 77.5 | 71.6 | 156 | 18% | L |
| | 23 | Zero streamflow days | 0.000 | 59.0 | 0.000 | 0.000 | −73% | H |
| | 24 | Base streamflow index | 0.022 | 0.000 | 0.08 | 0.028 | −93% | H |
| Group 3 | 25 | Minimum streamflow date | 152 | 151 | 128 | 174 | −28% | L |
| | 26 | Maximum streamflow date | 212 | 218 | 198 | 227 | 18% | L |
| Group 4 | 27 | Low pulse count | 3.00 | 2.00 | 1.92 | 6.00 | 6% | L |
| | 28 | Low pulse duration | 6.25 | 25.0 | 5.00 | 15.6 | −41% | M |
| | 29 | High pulse count | 4.00 | 2.00 | 3.00 | 6.00 | −31% | L |
| | 30 | High pulse duration | 7.00 | 17.0 | 5.46 | 14.4 | −54% | M |
| Group 5 | 31 | Rise rate | 0.662 | 0.382 | 0.447 | 0.991 | −54% | M |
| | 32 | Fall rate | −1.00 | −0.355 | −1.16 | −0.532 | −48% | M |
| | 33 | Number of reversals | 100 | 38.0 | 92.0 | 108 | −82% | H |

Remarks: The overall hydrological change was 54%, H means high change; M means moderate change; L means low change. Base streamflow index: 7-day minimum streamflow/mean streamflow for year. Rise/fall rates: median of all positive/negative differences between consecutive daily values. Number of reversals: by dividing the hydrologic record into "rising" and "falling" periods, the number of times that flow switches from one type of period to another. Low/high pulse: a day is classified as a pulse if it is greater than or less than a specified threshold. For this analysis, pulse thresholds are calculated using data in the pre-impact period only.

**Table 4.** Hydrological change degree of each group of IHA indicators.

| Hydrologic Change Degree of Each Group (%) | | | | | Degree of Overall Hydrologic Change (%) |
|---|---|---|---|---|---|
| **Group 1** | **Group 2** | **Group 3** | **Group 4** | **Group 5** | |
| 47 (M) | 66 (M) | 24 (L) | 37 (M) | 63 (M) | 54 (M) |

Remarks: M means moderate change; L means low change.

(1) Hydrological indicator selection

The 33 indicators of the IHA method had significant correlations with each other [58]. Correlation analysis was carried out on 33 indicator values, and the results are shown in Figure 5. The correlation between the indicators was quite high; for the range of absolute values of correlation coefficient, the maximum value was 0.999 and the minimum value was 0, with the average absolute value of the correlation coefficient being 0.332. The 1st to 6th indicators, as well as the 23rd to 33rd, had a large quantity of dispersion and outliers, but the 7th to 22nd indicators had little dispersion and fewer outliers. Indicators from 7 to 22 had a low degree of dispersion and few outliers. The PCA approach was used to assess each indication to reduce the degree of redundancy among the 33 indicators.

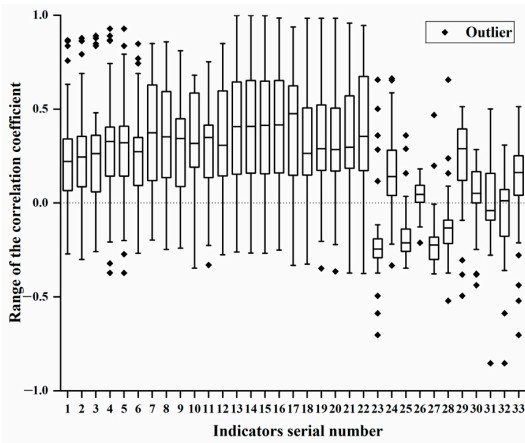

**Figure 5.** Correlation coefficients among the 33 IHA statistics.

The Kaiser-Guttman criterion was used to determine the number of primary components: the eigenvalue had to be larger than 1 and the cumulative contribution rate had to be at least 80%. The eigenvalues and cumulative contribution rates of the indicators of the 33 IHA methods are shown in Figure 6. The top nine principal component eigenvalues were all greater than 1, and the cumulative contribution rate was 85%, which met the Kaiser-Guttman requirement.

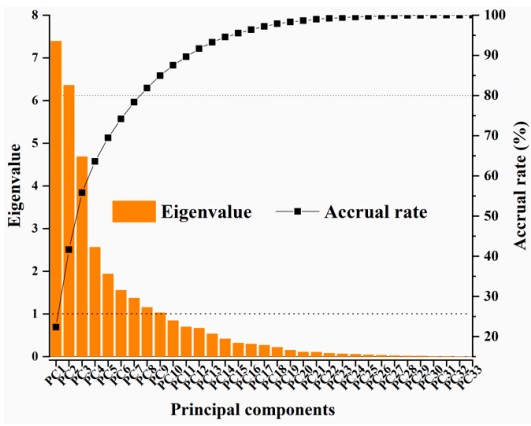

**Figure 6.** Eigenvalues and cumulative contribution rates for principal component analysis.

The load values of the top nine principal components are shown in Table 5, and the indicator with the largest load value among the top nine principal components was selected as the representative. A total of nine indicators were selected: 7-day maximum, median streamflow in January; 3-day minimum, fall rate, median streamflow in May; number of reversals, high pulse count, maximum streamflow date, median streamflow in June.

**Table 5.** Load value of each indicator of PC1 to PC9.

| IHA Indicators | Principal Component | | | | | | | | |
|---|---|---|---|---|---|---|---|---|---|
| | PC1 | PC2 | PC3 | PC4 | PC5 | PC6 | PC7 | PC8 | PC9 |
| Median streamflow in October | 0.045 | 0.897 | 0.011 | −0.027 | −0.063 | 0.006 | 0.131 | 0.001 | 0.126 |
| Median streamflow in November | 0.126 | 0.906 | −0.069 | −0.101 | −0.051 | 0.019 | 0.176 | 0.101 | 0.008 |
| Median streamflow in December | 0.082 | 0.936 | −0.052 | −0.047 | 0.134 | −0.079 | 0.013 | 0.133 | −0.036 |
| Median streamflow in January | 0.124 | 0.941 * | 0.064 | −0.117 | 0.099 | 0.023 | 0.018 | −0.056 | 0.035 |
| Median streamflow in February | 0.097 | 0.876 | 0.068 | −0.105 | 0.222 | 0.007 | −0.057 | −0.167 | −0.001 |
| Median streamflow in March | 0.051 | 0.772 | −0.050 | −0.053 | 0.402 | −0.121 | −0.092 | 0.094 | −0.199 |
| Median streamflow in April | 0.036 | 0.456 | −0.009 | −0.114 | 0.716 | −0.188 | −0.099 | 0.151 | −0.192 |
| Median streamflow in May | −0.072 | 0.041 | 0.191 | −0.031 | 0.872 * | 0.081 | 0.156 | −0.095 | 0.067 |
| Median streamflow in June | −0.084 | −0.061 | −0.036 | −0.108 | −0.041 | −0.109 | 0.036 | 0.094 | 0.861 * |
| Median streamflow in July | 0.211 | 0.182 | 0.148 | −0.005 | 0.209 | 0.093 | 0.291 | 0.733 | 0.123 |
| Median streamflow in August | 0.612 | 0.144 | −0.130 | −0.022 | −0.139 | 0.035 | 0.415 | −0.084 | 0.006 |
| Median streamflow in September | 0.399 | 0.234 | −0.054 | 0.019 | 0.044 | −0.012 | 0.709 | −0.221 | 0.169 |
| 1-day minimum | 0.111 | −0.027 | 0.971 | −0.076 | 0.093 | 0.006 | −0.034 | 0.074 | 0.023 |
| 3-day minimum | 0.092 | −0.001 | 0.977 * | −0.070 | 0.083 | 0.017 | −0.022 | 0.078 | 0.016 |
| 7-day minimum | 0.058 | 0.037 | 0.975 | −0.045 | 0.084 | 0.075 | −0.003 | 0.088 | 0.018 |
| 30-day minimum | 0.025 | 0.022 | 0.952 | −0.048 | 0.164 | 0.151 | 0.053 | 0.065 | 0.042 |
| 90-day minimum | 0.018 | 0.275 | 0.349 | −0.132 | 0.821 | 0.031 | 0.073 | 0.085 | −0.004 |
| 1-day maximum | 0.929 | 0.050 | 0.100 | 0.103 | −0.005 | −0.073 | −0.075 | −0.073 | −0.021 |
| 3-day maximum | 0.953 | 0.093 | −0.013 | 0.087 | 0.022 | −0.121 | −0.012 | −0.032 | −0.033 |
| 7-day maximum | 0.959 * | 0.055 | 0.041 | 0.042 | 0.047 | −0.147 | 0.029 | 0.003 | −0.053 |
| 30-day maximum | 0.939 | 0.065 | 0.023 | 0.104 | −0.015 | −0.073 | 0.157 | 0.062 | −0.034 |
| 90-day maximum | 0.859 | 0.158 | 0.051 | 0.093 | −0.004 | −0.001 | 0.371 | 0.097 | 0.131 |
| Zero streamflow days | −0.019 | −0.143 | −0.241 | 0.689 | −0.237 | −0.476 | 0.132 | −0.045 | −0.118 |
| Base streamflow index | −0.162 | −0.052 | 0.871 | −0.098 | 0.036 | 0.066 | −0.048 | −0.057 | −0.058 |
| Minimum streamflow date | −0.021 | −0.309 | −0.161 | 0.576 | 0.268 | 0.169 | −0.12 | −0.066 | −0.246 |
| Maximum streamflow date | 0.184 | 0.042 | −0.107 | 0.003 | 0.092 | −0.012 | 0.287 | −0.792 * | −0.011 |
| Low pulse count | −0.403 | −0.112 | −0.049 | −0.022 | −0.086 | 0.757 | −0.121 | 0.012 | −0.105 |
| Low pulse duration | −0.215 | −0.029 | −0.169 | 0.392 | −0.232 | −0.569 | −0.067 | 0.064 | 0.147 |
| High pulse count | 0.282 | 0.234 | 0.309 | −0.043 | 0.027 | 0.332 | −0.448 | −0.075 | 0.491 |
| High pulse duration | 0.288 | 0.009 | 0.056 | 0.124 | 0.212 | −0.317 | 0.729 * | 0.088 | −0.126 |
| Rise rate | 0.212 | −0.084 | −0.052 | 0.853 | −0.091 | −0.092 | 0.048 | 0.052 | −0.004 |
| Fall rate | −0.144 | 0.093 | 0.039 | −0.897 * | 0.098 | 0.243 | −0.068 | 0.019 | 0.007 |
| Number of reversals | −0.053 | −0.099 | 0.256 | −0.288 | −0.097 | 0.761 * | −0.178 | 0.151 | 0.068 |

Remarks: * indicates the highest load value in the corresponding principal component.

(2) Rationality analysis of preferred indicator

The degree of hydrological change estimated by the 33 IHA method indicators was compared to the total hydrological change of the selected nine indicators to see if they could effectively retain the information in the traditional IHA method indicators. Tables 4 and 6 illustrate the results. By comparison, total hydrological change of the original 33 indicators was 54%, and the overall hydrological change of the preferred indicators was 58%, both of which were modest changes, with an overall relative inaccuracy of hydrological change of 7%.

Because of the significant correlation among the indicators, they repeated the description of the degree of change in the overall hydrological evaluation, resulting in a small or large evaluation [21]. As a result, a correlation study of the nine indicators was conducted, and the findings are displayed in Figure 7. The result showed that the preferred indicators' correlation coefficient distribution was more concentrated compared to the original IHA

indicators' (Figure 5). The preferred value of the correlation coefficient of the nine indicators ranged from 0.839 to 0.228, with an average absolute value of 0.228; However, the absolute value of the correlation coefficient for the 33 IHA indicators ranged from 0 to 0.999, with an average value of 0.332. The correlation of the nine preferred indicators was much lower than that of the original 33. Therefore, the nine indicators chosen using principal component analysis are more suited to studying the Dawen River's hydrological variability.

**Table 6.** Results of preferred indicator evaluation.

| Serial Number | IHA Indicators | Hydrologic Alteration (%) |
|---|---|---|
| 4 | Median streamflow in January | 8% |
| 8 | Median streamflow in May | 80% |
| 9 | Median streamflow in June | 67% |
| 14 | 3-day minimum | 87% |
| 20 | 7-day maximum | 41% |
| 26 | Maximum streamflow date | 18% |
| 29 | High pulse count | 31% |
| 32 | Rise rate | 48% |
| 33 | Number of reversals | 82% |

Remarks: The overall hydrological change was 58%, which was a moderate change.

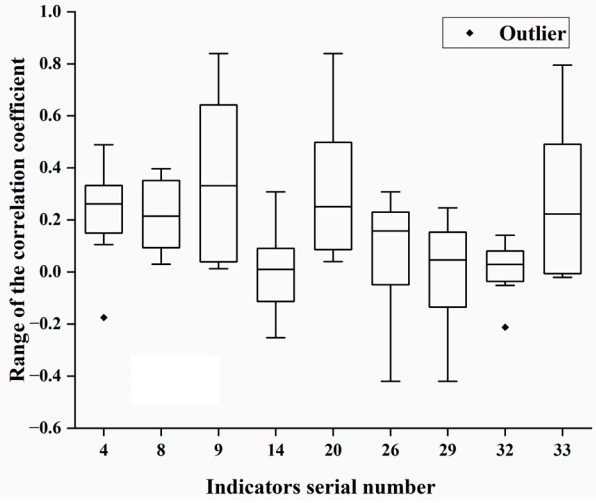

**Figure 7.** Correlation coefficient between 9 preferred indicators.

3.2.2. Analysis of Hydrological Regime Change

This research analyzed the indicators with greater variability and the preferred indicators in the IHA method before and after variation to explore the changes in the Dawen River's hydrological regime and clarify the Dawen River's variation law from many perspectives.

(1) Magnitude of monthly water conditions (Group 1)

The first group contained 12 indicators, among which were the hydrological change degree of two indicators: the median streamflow in May and median streamflow in June. They reached a high degree of change, and the hydrological change degree of the median streamflow in March, April, September, November, and December indicators was moderate. The hydrological changes of the median streamflow indices in January, February, July, August, and October were low. The monthly median streamflow in the two periods before and after the variation is shown in Figure 8. The monthly median streamflow following the mutation was lower than prior to the mutation. Precipitation showed an insignificant declining tendency, according to the interannual variation trend. The maximum median value of streamflow fell by 47% from 76.4 m³/s before mutation to 40.3 m³/s afterwards. The preferred indicators of median streamflow in January, May, and June were analyzed

in this paper, and the findings are displayed in Figure 9. The indicator of hydrological change degrees for the median streamflow in May and June were high. Moreover, these streamflows were smaller than that of January because the amount of water from interbasin water diversion in January was greater and irrigation for the basin's primary crops is concentrated in May and June. The reduction of streamflow will lead to the reduction of aquatic habitats, the decline of groundwater level, and the increase in surface water infiltration, thereby reducing the effective water content of soil and affecting terrestrial plants and terrestrial animals.

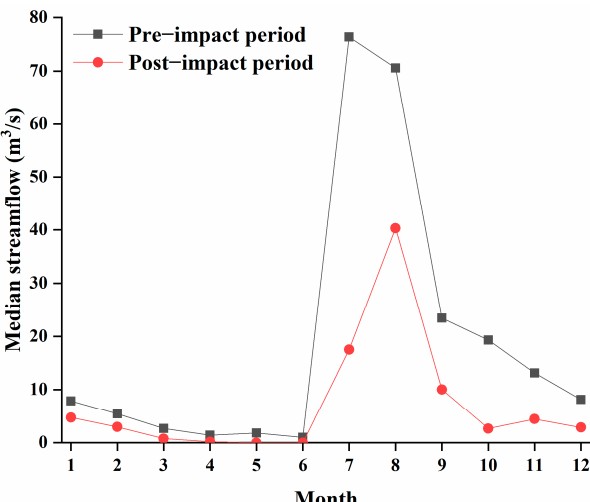

**Figure 8.** Monthly median streamflow.

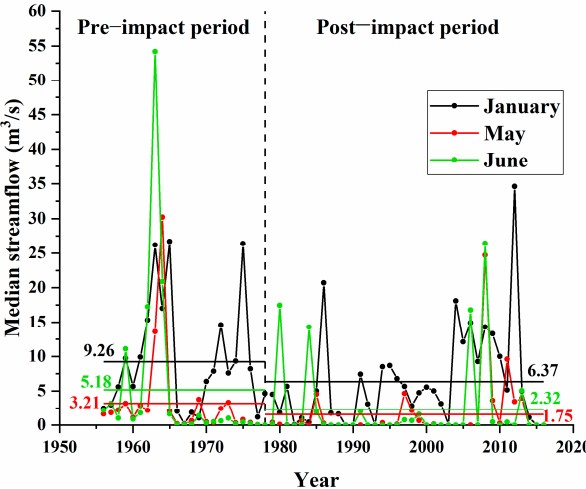

**Figure 9.** Median streamflow for January, May, and June.

(2)  Magnitude of annual extreme discharge events with different durations (Group 2)

There were nine indicators with moderate and high changes among the 12 indicators in this group, as indicated in Table 3. The indicator for zero streamflow days changed from 0 days before the mutation to 59 days afterwards. From 1956 to 1978, the average number of zero streamflow days was 4.17, but from 1979 to 2016, it was 96.5, with obvious differences. The indicator for the 90-day minimum changed from 1.28 $m^3$/s before the mutation to 0.245 $m^3$/s afterwards. It shows that the Daicun Dam Hydrological station on the Dawen River had a cut-out phenomenon following the mutation, and that it was severe, almost reaching a continuous cut-out for 90 days. As illustrated in Figure 10, the paper examined the two preferred indicators: 7-day maximum and 3-day minimum. These two preferred indicators were analyzed, and the results are shown in Figure 10. Compared to values before mutation, the average of the two indicators fell by 39% and 87%, respectively.

From the analysis of the 3-day minimum indicator, the number of days with no streamflow increased from 2 before the mutation to 29 afterwards, indicating that the Dawen River's hydrological variability is quite high and that there is a severe cut-off phenomenon. The morphological structure of river channels and natural habitat conditions, as well as the distribution of plant groups in lakes, ponds, and floodplains, would be affected by changes in this collection of indicators. Soil moisture stress in plants will occur due to the reduction in index values.

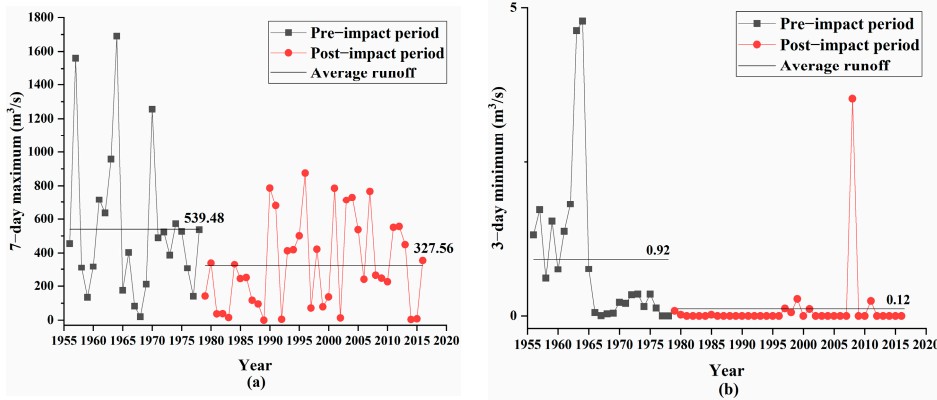

**Figure 10.** 7-day maximum (**a**) and 3-day minimum (**b**).

(3)  Timing of annual extreme water conditions (Group 3)

The two indicators in this group (minimum and maximum streamflow date) had small changes. The preferred indicator maximum streamflow date was examined, as illustrated in Figure 11. Because the Dawen River is seasonal, heavy rain and flooding are the key influencing factors. The inter-annual fluctuation of torrential rain and floods is not large since the climatic conditions in the basin do not change significantly; therefore, the preferred indicator did not change significantly before or after the variation. However, the years 1983 and 1992 were extremely dry. With no significant rainfall or floods, the rivers essentially stopped running, causing the peak flow to occur earlier. Changes in this group of indicators will lead to the evolution of biological life strategies and behavioral mechanisms.

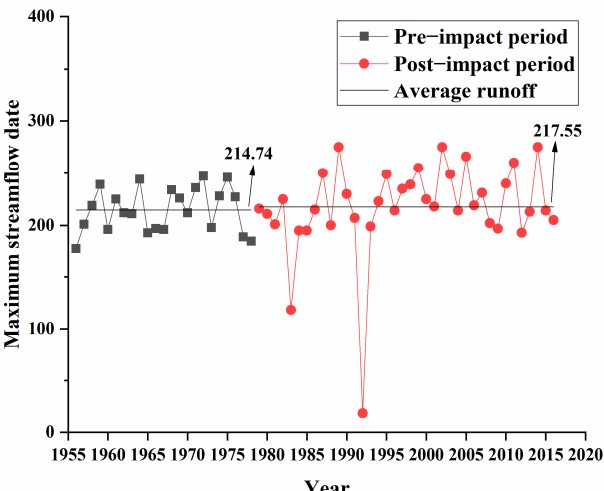

**Figure 11.** Maximum streamflow date.

(4)  Frequency and duration of high and low pulses (Group 4)

The changes in the two indicators of low pulse duration and high pulse duration were moderate, whereas the changes in low pulse count and high pulse count were low. The number of pulses dropped after the mutation, but the duration of each pulse increased, these changes perhaps leading to an increase in severe flow and increasing the difficulty

of rivers overcoming flood seasons. The preferred indicator in this paper was high pulse count, and its change is shown in Figure 12. This indicator's multi-year average decreased by roughly 50%, from 4.5 times before the mutation to 2.2 times after the mutation. Changes in this group of variables can lead to changes in nutrient and organic matter exchange between rivers and floodplains, as well as oxygen and water stress in plants.

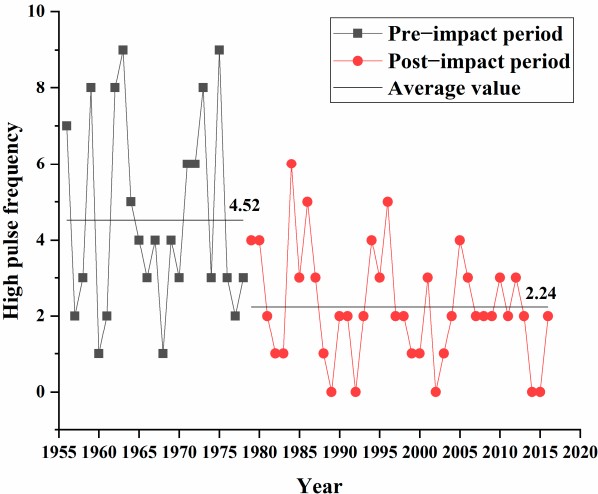

**Figure 12.** High pulse count.

(5)  Rate and frequency of water condition changes (Group 5)

In this group of indicators, the number of reversals showed a high degree of change, the rise rate and fall rate indicators showed moderate changes, and the above three indicators all showed a downward trend (Table 3). The authors of this paper analyzed the two preferred indicators, fall rate and number of reversals, as shown in Figure 13. The multi-year average of the number of reversal indicators fell from 99.5 before the mutation to 49.4 after the mutation, while the multi-year average of the fall rate indicator changed from −1.1 to −2.5. These two indicators differed dramatically before and after the variation, and the degree of dispersion rose, affecting the biological population of the river. The downward trend will lead to drought stress in plants and adversely affect the ecological environment.

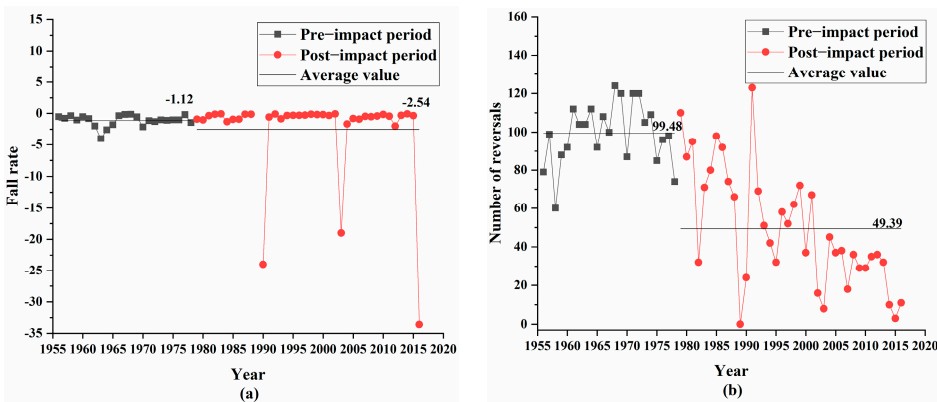

**Figure 13.** Fall rate (**a**) and number of reversals (**b**).

## 4. Discussion

This paper studied the evolution law of hydrological properties in the middle and upper portions of the Dawen River. The analysis discovered that both rainfall and runoff had a little decrease trend, with runoff's negative trend being more visible and volatile than rainfalls. The rainfall trend at most of the basin's 26 rainfall stations was downward; however, some stations had an upward tendency that was not clear. The mutation point of runoff was discovered to be 1978 during the investigation of the mutation point. The

time series was separated into two sections by this mutation point: the pre-influence stage and the post-influence stage. The overall hydrological change degree of the runoff in the Dawen River Basin was 54%, which was a moderate change, according to the IHA/RVA method. This study used the PCA method to tackle the problem of too many indicators in the IHA/RVA method and selected nine more representative indicators. These nine indicators were more suitable for analyzing changes in the Dawen River's hydrological regime, according to rationality analysis. Since the 1980s, the Daicun Dam Hydrological Station has frequently cut off water flow, with the cut-off time continually increasing; this phenomenon poses a threat to agricultural water use. Some sections are unable to maintain ecological flow, and pollution is becoming increasingly severe. The biology, habitat, and structure of the Dawen River have all been irreversibly impacted by changes in its hydrological regime. Determining the factors responsible for the hydrological variation has attracted the attention of experts and scholars. Climate change mainly affects runoff through precipitation and temperature [59], and in recent years, the precipitation in the Dawen River Basin has shown a downward trend, but it is not obvious. The decrease in precipitation, however, was not enough to account for the height change in the Dawen River or even the disconnection phenomenon. Evaporation in the Dawen River Basin likewise showed a declining trend, but this one was obvious [60]. Although the decrease in evaporation resulted in increased runoff, the measurements of the Dawen River showed it to be on the decline. Therefore, the main influencing factors of runoff change in the Dawen River are not climatic but human [61]. Especially since the 1970s and 1980s, the Dawen River has been the site of a large number of flood control and water-storage projects and land development, which seriously affected the underlying surface conditions causing the hydrological variation.

According to statistics, 20 large and medium-sized reservoirs and 438 small reservoirs were built in the basin above the Dawen River Daicun Dam Hydrological Station in the 1960s and 1970s. By 1978, the total capacity of the new reservoirs was 1.14 billion m$^3$. As shown in Table 7 and Figure 14, these projects brought great irrigation and economic benefits to local residents, but they also greatly changed the river's ecological and hydrological conditions. From 1979 to 2016, construction continued as three medium-sized reservoirs and 110 small reservoirs were built, but they had a total storage capacity of only 140 million m$^3$. In the mountainous areas in the north Dawen River Basin, soil and water loss is very serious, so in the river's upper reaches, a large amount of soil and water conservation work has recently been carried out: construction of terraces, conversion of cropland to forest and grassland, and digging ecological ditches [62,63]. Soil and water conservation measures, however, reduce the amount of runoff, resulting in river–runoff disconnection, which affects the environment of the whole basin [64]. The underlying surface condition of the Dawen River Basin has significantly changed: as water storage capacity increased, runoff decreased. In addition, with the increase in sand mining in the middle and lower reaches, the original river channel widened, increasing riverbed water storage and reducing runoff. Therefore, the main influencing factors of hydrological variation in the Dawen River are human, which is consistent with the conclusions concerning the main influencing factors of hydrological variation in the Yellow River [58,65].

**Table 7.** Comparison of the number of new reservoirs before and after the mutation.

| | New Construction from 1956 to 1978 | New Construction from 1979 to 2016 |
|---|---|---|
| Large reservoirs | 2 | 0 |
| Medium reservoirs | 18 | 3 |
| Small reservoirs | 542 | 110 |
| Total storage capacity ($10^4$ m$^3$) | 114,183.17 | 14,253.78 |

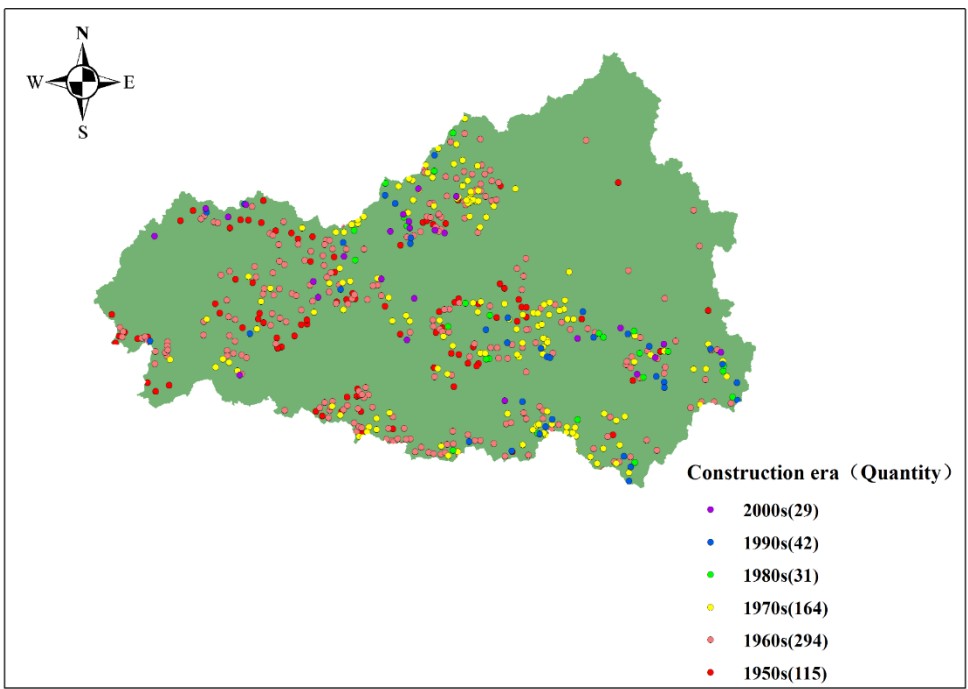

**Figure 14.** Distribution of new reservoirs in the Dawen River Basin during the study period.

In recent years, river runoff in northern China has decreased significantly. The Yellow, Songhua, and Haihe rivers, among others, have become typically water deficient. Studies have shown that more than 50% of the reduced runoff is caused by human activity [66], which has attracted wide attention from people of all walks of life. According to Li et al. [67], soil and water conservation measures in the Wuding River, a tributary of the Yellow River, were the most important variables affecting reduced river discharge from 1976 to 1997. Zhao et al. [68] investigated flow fluctuations in the Yellow River Basin's middle reaches and evaluated the impact of climate change and human activity. The investigation found that water conservation projects, soil and water conservation, and water consumption were the key factors that caused the dramatic decline in annual runoff over the past 60 years. The conclusions of this study are similar with the findings of the previous studies, which suggest that the impact of human activities on runoff is gradually rising.

**5. Conclusions**

The study of river hydrological variability is the basis for the optimal allocation of river water resources and the study of water ecological protection. In this paper, the evolution characteristics of the hydrological parameters in the Dawen River were studied using a variety of statistical approaches. The method of combining PCA and IHA was used to study the changing law of the hydrological regime of Dawen River. At the same time, the influence of hydrological regime variation on river ecological environment and the influencing factors of hydrological variation were analyzed. In summary, the overall trend of the Dawen River rainfall and runoff was downward, although the negative tendency was more pronounced. The overall hydrological variability of the Dawen River was moderately variable, and since the 1980s, the Dawen River has experienced severe dry-flow phenomena. This not only has a certain impact on the production and domestic water in the region, but also has an irreversible impact on the river's biology, ecological environment, and structure. In addition, the construction of a large number of water storage projects is the main influencing factor of the hydrological variation of the Dawen River. Therefore, in order to alleviate the shortage of water resources in this area and improve the ecological environment, it is suggested that the water resource allocation of the Dawen River be optimized and that the ecological environment of the river be improved by means of joint scheduling of reservoirs. This achievement has important practical significance and

theoretical value for the rational development and utilization of water resources and ecological environment protection.

**Author Contributions:** Conceptualization, Z.Z., Y.W. and J.L. (Jingqiang Liu); methodology, Y.L. and L.Z.; software, L.Z.; investigation, Y.L. and L.Z.; data curation, L.H. and J.L. (Jingqiang Liu); writing—original draft preparation, Y.L., L.Z. and L.H.; writing—review and editing, Y.L., L.Z., L.H. and J.L. (Jianxin Li); funding acquisition, L.H. All authors have read and agreed to the published version of the manuscript.

**Funding:** This research was funded by the Open Research Fund for State Key Laboratory of Simulation and Regulation of Water Cycle in River Basin, China Institute of Water Resources and Hydropower Research (No.: IWHR-SKL-202219); Research on the current situation and safeguard measures of ecological flow in the main stream of the Dawen River (No.: tasswzx2022_077); the Water Source Comprehensive Supervision Project (No.: WE0149B052020); and Shandong Province Graduate Education Quality Improvement Plan (No.: SDYAL19128). Informed consent was obtained from all subjects involved in the study.

**Institutional Review Board Statement:** Not applicable.

**Informed Consent Statement:** Not applicable.

**Data Availability Statement:** The data sets used and/or analyzed during the current study are available from the corresponding author upon reasonable request.

**Conflicts of Interest:** The authors declare no conflict of interest.

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
