# Peer review of "Research on the Hydrological Variation Law of the Dawen River, a Tributary of the Lower Yellow River"

_agronomy, doi:10.3390/agronomy12071719_

Round 1
Reviewer 1 Report
· The content presented in the paper corresponds to the Water Journal rather than the Agronomy Journal.
· The abstract should be carefully rewritten. In the current form it is too long and the maximum length of the abstract according to the instructions for authors is 200 words. It should be presented according to the scheme background, methods, results and main conclusions.
· It should be emphasized clearly what is the main objective of this paper, in the present form it is formulated very generally " The major goal of this research was to investigate the Dawen River's precipitation and streamflow evolution features".
· The paper should clearly state what scientific problem Authors are trying to solve. State the research hypothesis and then verify it.
· Indicate what is the new element of the research presented in this paper.
· Figure 1: Correlate the colors of Shandong Province in the figures. Explain what the abbreviation DEM means and complete the units. Hydrological station is not visible.
· Complete the symbols in the formulas and, if possible, standardize them in the presented formulas.
· In the methodology, it should be explained what exactly MK mutation test is.
· Also, when describing the IHA/RVA method, all the symbols used in the formulas should be given.
· I believe that the PCA method is presented in too much detail. Consideration should be given to rewriting this description.
· Section 3.1.1 should not start with a figure.
· In figure 1 the Authors have presented all meteorological stations. One should clarify which particular station is concerned by the analysis presented in figure 2. Or change the description of the figure.
· In figure 3 precipitation should be given with an accuracy of 1 mm.
· There is no broader discussion of the results in the paper. I think it should be a separate section of the paper.
· There should be a clearer indication of what the Authors mean by this „The research results in this paper can be used as a scientific foundation for improving water resource allocation in the Dawen River Basin”.
· Indicate in the paper how the presented studies are related to the profile of the Agronomy Journal. It is worth highlighting in the introduction and discussion sections.
Author Response
Dear Editor, Dear reviewers
We would like to thank you very much for your valuable comments and good suggestions that greatly helped to improve our manuscript. Thank you very much for your time and efforts. Based on the comments and suggestions, we have revised the manuscript accordingly. And we have uploaded a version of the revised manuscript with all the changes highlighted by using the track changes mode in MS word and a version of the revised manuscript with all changes accepted.
Appended to this letter is our point by point response to the comments raised by the reviewers,please see the attachment. The comments are reproduced and our responses are given directly afterward in a different color (red). The line number in Reply refers to the revised manuscript with track changes.
We appreciate for Editor and Reviewers' warm work earnestly, and hope that the correction will meet with approval.
Once again, thank you and all the reviewers for the advice and consideration.
Sincerely,
Long Zhao
College of Water Conservancy and Civil Engineering, Shandong Agricultural University
Tai’ an, Shandong Province, China
E-mail: zlllong@gmail.com

Reviewer 2 Report
I have read with interest your paper on Research on the hydrological variation law of the Dawen River, a tributary of the lower Yellow River. Presented problem could be interesting, however, I have a few major as well as minor comments.
First of all, I found it very hard to follow the language of the manuscript. It is not a matter of a few small to moderate language errors that are typical in papers not written by native speakers. Some parts are basically not comprehensible due to poor grammar and style. As a result many conclusions and argumentation are simplified too much, look illogical or sound incomprehensibly. Moreover, authors should avoid emotional and colloquial words; for example: “runoff fell dramatically”, “to examine the situation of the 33 indicators” etc. Therefore I strongly suggest that the manuscript should be proofread by a professional translator / native speaker and resubmit.
In chapter ‘Results and discussion’ where you analyse the groups of parameters, their physical/physico-geographical/hydrological/environmental interpretation is highly expected. You just deal with numbers without environment. In chapter 3.3, which should have a discussion I suppose, lacks linking to results of the paper and comparison to other results. There is a few vague remarks not specifically connected with paper results.
The ‘Conclusion’ chapter is hardly connected with main body, poor in scope and contains ‘Summary’ not ‘Conclusion’. There is no reference to work goals in relation to positive and negative results of the analysis as well as wider significance of the results. Moreover, in conclusion (1) we read about ‘overall negative trend in rainfall and runoff’ whereas in chapter 3.1.1 ONLY ONE case of statistically significant trend was proved. Such kind of scientific facts manipulation casts in doubt the reliability of presented results. Please check manuscript carefully if you want avoid rejecting and be aware that lack of significant trends as worth conclusion as their existence.
I also have some methodical objections:
page 10-11 fig, 3 – if precipitation trends are statistically insignificant, that means they DO NOT EXIST. Lack of significance concerns especially the SIGN of tendency (+/-). It is a serious methodical mistake to interpret statistically insignificant signs. Instead of this, paper should have a clear conclusion about overall lack of statistically significant trends and try to explain its determinants.
The break point on double-mass curve is not convincing (Fig. 4). It looks like parallel shift after a few years with higher precipitation in 70s. (in the middle of 60s. as well). Therefore, it does not look as a beginning of significant anthropogenic impact. Separation of these two time periods should be proved by statistical tests, e.g. Independent Samples t Test, Brown-Forsythe test of homogeneity of variance for two or more populations, nonparametric Mann-Whitney U Test.
Tab. 3 – Detailed information about definition and calculation way is needed. It concerns especially: base streamflow index, low/high pulse count and duration, rise/fall rate, number of reversals.
What was the particular criteria for dividing results into low/middle/high ranges?
l. 365-373 – Average correlation about 0.332 is not quite high. You forgot about the coefficients form range -1.0 – 0.0 (l. 368). What exactly proofs this graph? What about interpretation the indicators with coefficients placed below zero?
To sum up, the manuscript should be seriously and widely revised.
Author Response

(The authors gave the same response as above.)
